# *MalaSelect*: A Selective Culture Medium for *Malassezia* Species

**DOI:** 10.3390/jof7100824

**Published:** 2021-10-01

**Authors:** Abdourahim Abdillah, Stéphane Ranque

**Affiliations:** 1Aix Marseille Université, Assistance Publique-Hôpitaux de Marseille (AP-HM), Institut de Recherche pour le Développement (IRD), Service de Santé des Armées (SSA), VITROME: Vecteurs—Infections Tropicales et Méditerranéennes, 19-21 Boulevard Jean Moulin, 13005 Marseille, France; abdourahim15@live.fr; 2IHU Méditerranée Infection, 19-21 Boulevard Jean Moulin, 13005 Marseille, France

**Keywords:** *Malassezia*, selective culture medium, rapamycin, isolation, polymicrobial samples

## Abstract

*Malassezia* species are fastidious and slow-growing yeasts in which isolation from polymicrobial samples is hampered by fast-growing microorganisms. *Malassezia* selective culture media are needed. Although cycloheximide is often used, some fungi, including the chief human commensal *Candida albicans*, are resistant to this compound. This study aimed to test whether the macrolide rapamycin could be used in combination with cycloheximide to develop a *Malassezia*-selective culture medium. Rapamycin susceptibility testing was performed via microdilution assays in modified Dixon against two *M. furfur* and five *Candida* spp. The MIC was the lowest concentration that reduced growth by a minimum of 90%. Rapamycin ± cycloheximide 500 mg/L was also added to FastFung solid, and yeast suspensions were inoculated and incubated for 72 h. Rapamycin MICs for *Candida* spp. ranged from 0.5 to 2 mg/L, except for *C. krusei*, for which the MIC was >32 mg/L. *M. furfur* stains were rapamycin-resistant. Rapamycin and cycloheximide supplementation of the FastFung medium effectively inhibited the growth of non-*Malassezia* yeast, including cycloheximide-resistant *C. albicans* and *C. tropicalis*. Based on our findings, this “*MalaSelect*” medium should be further evaluated on polymicrobial samples for *Malassezia* isolation and culture.

## 1. Introduction

The *Malassezia* genus comprises 18 species that are lipid-dependent yeast commensals on human skin and other warm-blooded vertebrates [1]. Under certain circumstances (e.g., increased humidity and temperature), these yeasts cause common human skin diseases, including pityriasis versicolor, seborrheic dermatitis and folliculitis [2]. They can also cause severe bloodstream infections in neonates or immunocompromised patients hospitalized in intensive care with lipidic parenteral nutrition [3,4,5,6]. Their role in atopic dermatitis and psoriasis was also reported [7,8]. However, data on the interactions between *Malassezia* species and their hosts remain scarce, especially in humans. This knowledge gap has been widened because the routine use of culture media on which these lipid-dependent yeasts can be cultivated is limited to a few specialized laboratories. In the literature, a variety of lipid-enriched culture media, including Dixon and Leeming–Notman agar and their modified versions, are effective for *Malassezia* spp. cultivation [9,10,11]. However, these culture media are not selective, in that they also allow the growth of a variety of other fungi, including *Candida* spp. and molds [12,13,14,15]. Relatively slow-growing *Malassezia* spp. can remain undetected, overtaken by faster growing and sometimes also more abundant fungal species present in the samples. Therefore, the use of antimycotic compounds without effect on *Malassezia* spp. seems to be an option for the development of a selective culture medium. Among these types of compounds, cycloheximide is widely used in culture media aimed at isolating and cultivating *Malassezia* spp. from clinical samples [16,17,18,19]. However, some fungi, especially in the genus *Candida* (e.g., the most common human opportunistic pathogen *C. albicans*), are resistant to cycloheximide and might thus interfere with *Malassezia* isolation. The development of a *Malassezia*-specific and efficient culture medium is crucial in order to be able to selectively isolate *Malassezia* spp. from polymicrobial clinical specimens, as these yeast species are detected at relatively higher frequencies by using culture-independent methods in complex polymicrobial niches such as the respiratory and digestive tracts [20,21,22,23,24].

Vézina et al. discovered rapamycin in 1975 [25]. This is a secondary metabolite produced by *Streptomyces hygroscopicus* and has antifungal activity, especially against *Candida albicans*. Rapamycin is now commonly used as an immunosuppressive drug [26]. Although its antifungal activity against molds or other yeast species is well known [27,28], the potential activity of rapamycin against *Malassezia* spp. is poorly studied. However, it was recently reported that *M. furfur* and *M. sympodialis* are not sensitive to rapamycin [29]. The aim of this present study was to assess the in vitro antifungal activity of rapamycin against *Malassezia* spp. and *Candida* spp. by using a broth microdilution method, and to test whether a solid culture medium supplemented with rapamycin and cycloheximide might allow the growth of *Malassezia* spp., while inhibiting the growth of *Candida* spp.

## 2. Materials and Methods

### 2.1. Strains

A total of 3 *Malassezia* species, obtained from the Belgian Co-ordinated Collections of Micro-organisms/Institute of Hygiene and Epidemiology (BCCM/IHEM, Sciensano, Brussels, Belgium), including *M. furfur*, *M. sympodialis* and *M. pachydermatis*, were used to test the antifungal activity of rapamycin. *Malassezia* species were maintained at 30 °C and subcultured for 5 days before testing on FastFung medium [30,31], composed per liter [pH 6] of 43 g of Schaedler agar, 20 g of peptone, 10 g of glucose, 5 g of ox-bile, 10 g of malt extract, 2 mL oleic acid, 2.5 mL glycerol and 5 mL of Tween 60 (all from Sigma-Aldrich, Saint-Quentin Fallavier, France). Isolates of *C. albicans*, *C. glabrata*, *C. parapsilosis* and *C. tropicalis* and the *C. krusei* ATCC 6258 strain were also used. The *Candida* species were subcultured at 30 °C for 3 days in Sabouraud medium before testing.

### 2.2. Broth Microdilution

Rapamycin solutions (Sigma-Aldrich, Saint-Quentin-Fallavier, France, Ref. S-015-1mL) at 1 mg/mL in acetonitrile were obtained and stored at −80 °C until use. The rapamycin concentration gradient tested ranged from 0.0625 to 32 mg/L. Broth microdilution method was performed by using modified Dixon broth (3.6% malt extract, 0.6% peptone, 2% ox-bile, 1% Tween 40, 0.2% glycerol, and 0.2% oleic acid, buffered at pH 6) (all from Sigma-Aldrich Saint-Quentin-Fallavier, France). Yeast inoculum suspensions of *M. furfur* and *Candida* spp. were prepared in 2 mL sterile saline solution (0.85% NaCl) and standardized spectrophotometrically at 0.5 McFarland (10^6^ colony-forming units [CFU]/mL). These suspensions were diluted 1:10 in sterile distilled water, and a total of 100 µL of the final dilution was transferred into a 96-well microtiter plate containing 100 µL of the medium to achieve a final concentration of 0.5–2.5 × 10^5^ CFU/mL as recommended by EUCAST. Each assay was tested in duplicate. The microtiter plates were incubated at 30 °C and visually read after 24 and 48 h of incubation. The growth of each strain at various rapamycin concentrations, as well as a positive control cultured in rapamycin-free medium, was recorded. The MIC of each strain was defined as the lowest concentration that reduces growth by a minimum of 90% when compared to the control growth.

### 2.3. Growth Testing on Agar

FastFung medium supplemented with 500 mg/L cycloheximide (CliniSciences, Nanterre, France) was prepared and sterilized by autoclaving at 121 °C for 30 min. After cooling to approximately 56 °C, FastFung medium was partitioned into two equal volumes, and a solution of rapamycin was added to one volume of the FastFung medium. The two media were then distributed in sterile Petri dishes. *Malassezia* spp. and Candida spp. yeast suspensions at 10^6^ CFU/mL were prepared and further diluted at 10^5^, 10^4^ and 10^3^ in sterile distilled water. For each dilution, 20 µL was plated onto FastFung medium with or without rapamycin. Sterile distilled water was also plated in FastFung medium as a negative control. All agar plates were incubated aerobically at 30 °C and examined daily for 3 days.

## 3. Results

Rapamycin antifungal activity was evaluated in Dixon liquid medium against 2 *M. furfur* and 5 *Candida* strains, including *C. albicans*, *C. glabrata*, *C. tropicalis*, *C. parapsilosis* and *C. krusei* ATCC 6258. The MIC results are summarized in Table 1. Rapamycin exhibited low MIC values against *Candida* spp., except for *C. krusei* ATCC 6258, whose MIC were ≥32 mg/L. MIC after 24 h of incubation were 1- to 2-fold dilutions lower than those recorded after 48 h of incubation (Table 1). Slow-growing *M. furfur* required 48 h reading time. Both *M. furfur* strains exhibited high rapamycin MIC ≥ 32 mg/L, suggesting that rapamycin had no significant effect against *Malassezia* spp. After determination of rapamycin MIC values against *M. furfur* and *Candida* spp., we assessed the growth of these isolates on FastFung medium plates supplemented or not with 2 mg/L rapamycin. The plates were examined daily for 3 days. No growth was observed on FastFung medium with cycloheximide, supplemented or not with rapamycin for *C. glabrata*, *C. parapsilosis* and *C. krusei* ATCC 6258 (Figure 1). Cycloheximide (500 mg/L) did not inhibit *C. albicans* and *C. tropicalis* in FastFung medium without rapamycin. In FastFung medium supplemented with both cycloheximide and rapamycin, *C. albicans* and *C. tropicalis* did not grow after 2 to 3 days of incubation. The growth of *M. furfur*, *M. sympodialis* and *M. pachydermatis* was not altered when these yeasts were cultured in FastFung medium containing cycloheximide, supplemented or not with rapamycin (Figure 1).

## 4. Discussion

The development of selective culture media is an important challenge in medical mycology for isolation and culture of pathogens from clinical specimens. The use of antimicrobial agents remains the main strategy for the inhibition of undesirable microorganisms in culture. Here, we evaluated the in vitro antifungal activity of rapamycin against 2 *M. furfur* and 5 *Candida* spp. and found that *M. furfur* is resistant, with MIC values ≥ 32 mg/L (Table 1). However, rapamycin showed higher activity against *Candida* spp., with MIC values ranging from 0.5 to 2 mg/L, except *C. krusei* ATCC 6258 (Table 1). This higher activity of rapamycin against *Candida* spp., especially *C. albicans,* is consistent with results reported in the literature [25,27,28]. MIC values ≥ 32 mg/L recorded for *C. krusei* ATCC 6258 deserve further investigation to determine if rapamycin does not affect *C. krusei* species or whether our strain is resistant.

Growth testing on agar showed that cycloheximide and rapamycin have dual negative effects against *Candida* spp. in culture. Cycloheximide inhibited the growth of *C. glabrata*, *C. parapsilosis* and *C. krusei* ATCC 6258, whereas *C. albicans* and *C. tropicalis* were able to grow in FastFung medium supplemented with cycloheximide (Figure 1). The effects of cycloheximide against certain *Candida* spp. are already known [12,14,15]. By adding rapamycin, we were able to inhibit all *Candida* spp. tested, as no growth was observed in culture (Figure 1). No effect of cycloheximide against *Malassezia* species was observed, which is in agreement with the literature [17,18].

One of the most striking findings was the resistance of *Malassezia* spp. to rapamycin. Both strains of *M. furfur* showed resistance, with MIC values ≥ 32 mg/L (Table 1). These results were confirmed by testing *M. furfur*, *M. sympodialis* and *M. pachydermatis* in FastFung medium supplemented with rapamycin (Figure 1). Our findings are particularly interesting, as rapamycin inhibits a broad spectrum of fungi, including filamentous fungi and yeasts [28]. These results show that rapamycin can be a good candidate in selective agents. For example, it is well known that *Candida* spp. grow quickly, whereas *Malassezia* spp. grow slowly. Rapamycin allows the growth of *Malassezia* spp. by limiting the growth of *Candida* spp. when samples with multiple fungal species are analyzed. On the other hand, rapamycin can prevent contamination during subculturing. Contamination problems are frequent in slow-growing fungi [32].

## 5. Conclusions

Based on our findings, we propose the use of culture media such as FastFung, supplemented with cycloheximide and antibiotics, including rapamycin, for isolation and culture of *Malassezia* species from polymicrobial samples. There is no standardization for susceptibility testing of Malassezia, and only a very limited number of strains have been tested, so this result warrants further investigation. In the future, it will be interesting to test this selective medium with clinical specimens of varied origins, including stool.

## Figures and Tables

**Figure 1 jof-07-00824-f001:**
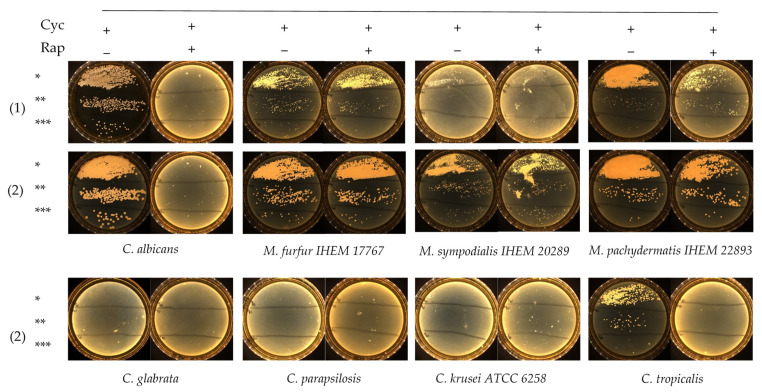
Growth testing via serial inocula dilution of *Candida* spp. and *Malassezia* spp. on the FastFung medium without (−) or with (+) 2 mg/L rapamycin (Rap) and 500 mg/L cycloheximide (Cyc). Culture at 48 h (1) and 72 h (2) of incubation. * 10^5^ CFU/mL, ** 10^4^ CFU/mL and *** 10^3^ CFU/mL.

**Table 1 jof-07-00824-t001:** Rapamycin in vitro susceptibility testing against *M. furfur* and *Candida* spp.

Strains	Rapamycin MIC (mg/L)	
	24 h	48 h
*C. albicans*	0.25	0.5
*C. glabrata*	0.5	2.0
*C. tropicalis*	0.25	0.5
*C. parapsilosis*	0.5	1.0
*C. krusei* ATCC 6258	32.0	>32.0
*M. furfur* IHEM 17767	-	>32.0
*M. furfur* IHEM 19320	-	>32.0

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
