# Peer review of "MalaSelect: A Selective Culture Medium for Malassezia Species"

_jof, 2021, doi:10.3390/jof7100824_

Round 1

Reviewer 1 Report

Dear authors,

although the preliminary results of your study are interesting, this manuscript contains some flaws.

1) Susceptibility testing has not been performed how it should be, meaning that final concentration of acetonitrile was way too high.

2) Only a very limited number of strains (and Malassezia spp.) underwent been susceptibility testing of rapamycin. Additionally, it has not been shown that the majority of C. albicans and C. tropicalis strains are susceptible to rapamycin.

3) Your recommendation is not supported by your results, as you have not tested polymicrobial samples.

Abstract

Line 9: Whose is not the correct relative pronoun in that kind of sentence.

Line 10-12: I don’t understand what the authors want to say here.

Line 15: What is Dixon? The readers need to know how many strains of each species have been tested.

Line 17 What is FastFung solid?

Line 18: I am no native English speaker, but think whose is again not correct here.

Line 21-22: Based on your results you cannot recommend the use of your medium for polymicrobial samples, as you have not tested them! This medium could be further evaluated on polymicrobial samples whether it works, or not.

Introduction

Line 29: Please delete etc. or define etc.

Line 29-31: Is it correct that these fungi can cause bloodstream infections in neonates or immunocompromised patients when they are hospitalized in intensive care with parenteral nutrition and if they don’t receive parenteral nutrition, the can’t cause bloodstream infections? If no, please correct the meaning of the sentence.

Line 35: Again, what is Dixon? Is it liquid or solid?

Line 36: Please write microorganisms in italic throughout the whole manuscript.

Line 37: Candida spp. are yeasts. Candida spp. and moulds or yeasts in including Candida spp. and moulds?

Line 39. Please delete the word yeast.

Line 40. Should read “seem”.

Line 43. Should read “cycloheximide”.

Line 51: This “is” a …, Streptomyces hygroscopicus “and” has.. Please let the English of this manuscript be corrected by native English speaker. I stop correcting errors at this point.

Materials and Methods

Line 64: How is it possible if you have 3 strains from 3 different species to do susceptibility testing of rapamycin on 2 M. furfur strains?

Line 67: What means maintained? you stored them at 30°C in this medium? How where the strains subcultured before the experiments? Time of growth? Growth conditions? How did you culture the Candida strains?

Line Line 74/76: Why do you change the unit from mg/ml to mg/L?

Line 75-76: It is more than unclear how these dilutions have been performed. If rapamyin at a concentration of 1000µg/ml in cytotoxic acetonitrile was used, the final concentration of acetonitrile in the well containing 32µg/ml would be >3%!! This is way too much to run to run susceptibility testing without cytotoxicity of acetonitrile. Why didn’t you use DMSO as a solvent? Up to 200mg are soluble per ml. With DMSO it would easily be possible to stay under the 0.5% v/v final concentration of DMSO in each well.

It is not clear if all wells contained to same concentration of acetonitrile, including the growth control.

There exist protocols of modified EUCAST susceptibly testing of Malassezia spp. (10.1128/JCM.00338-17) which could have been used.

Line 80: Standardized to what? 0.5 McFarland?

Line 87-89: How is it possible to read an inhibition of 90% by eye? When I read MIC with 90% of inhibition I use a photometer.

Line 94: How much rapamycin was added to the medium?

Line 95: Did all petri dishes contain the same amount of medium?

Results

Line 104-105: Did you use the same concentrations of acetonitrile in the other paper?

Line 108: It’s recommended to read MIC of Candida species after 24h of incubation.

Line 109-110: Required 48h of reading time or growth time?

Line 111: That’s wrong. It is not suggesting that rapamycin has no activity against Malassezia spp. as you only tested 2 M. furfur strains! What about the other 17 species from the genus Malassezia? Why were the other species of your collection not tested?

Line 119: By testing 1 stain of C. albicans and 1 C. tropicalis you did not show that most strains of this species are susceptible to rapamycin.

Discussion

Line 133: Issue or challenge?

Line 137: Limited to 2 tested isolates of M. furfur.

Line 157: These results shows that rapamycin could be..

Line 159-161: When have you done this? I have only seen incubation with Candida spp. or Malassezia spp.

Line 163-166: You can propose to use your medium on polymicrobial samples, but you have not shown that it works. To my opinion your results warrant further investigation on polymicrobial samples, after it has been tested if the other clinically relevant Malassezia have been tested for their susceptibility to rapamycin.

Conclusion

Please delete conclusion or line 163-166 of the discussion.

Author Response

Although the preliminary results of your study are interesting, this manuscript contains some flaws.

  • Susceptibility testing has not been performed how it should be, meaning that final concentration of acetonitrile was way too high.

  • Rapamycin was supplied by the manufacturer (Sigma) in solution in acetonitrile (as solvent). We did not have to worry about the concentration of acetonitrile because the yeasts grew very well in our culture conditions. The only limiting factor for growth was rapamycin.

  • Only a very limited number of strains (and Malassezia) underwent been susceptibility testing of rapamycin. Additionally, it has not been shown that the majority of C. albicans and C. tropicalis strains are susceptible to rapamycin.

  • It is true that the rapamycin susceptibility testing was not performed on several strains of Malassezia. The susceptibility testing with 2 strains of Malassezia was a proof of concept. However, we further confirmed the results with at least 3 Malassezia species on solid media.
  • The effect of rapamycin against Candida was already known in the literature (see references 25, 27 and 28); therefore, we did not have to test many strains.

3) Your recommendation is not supported by your results, as you have not tested polymicrobial samples.

  • It is true that we did not test polymicrobial samples, but this is a recommendation for future studies. Moreover, we have a study in progress in which we are successfully testing this medium for the isolation of Malassezia from stool samples.

Abstract

Line 9: Whose is not the correct relative pronoun in that kind of sentence.

  • See correction line 9

Line 10-12: I don’t understand what the authors want to say here.

  • See reformulation line 10-12

Line 15: What is Dixon? The readers need to know how many strains of each species have been tested.

  • Dixon is a culture medium for Malassezia species, and this medium is known in the literature. Due to the number of words limit, it is not possible to detail all the strains of each species in the abstract.

Line 17 What is FastFung solid?

  • This is a culture medium developed in our laboratory (see reference 30-31)

Line 18: I am no native English speaker, but think whose is again not correct here.

  • See correction line 18-19

Line 21-22: Based on your results you cannot recommend the use of your medium for polymicrobial samples, as you have not tested them! This medium could be further evaluated on polymicrobial samples whether it works, or not.

  • See reformulation line 21-22

Introduction

Line 29: Please delete etc. or define etc.

  • See correction line 29

Line 29-31: Is it correct that these fungi can cause bloodstream infections in neonates or immunocompromised patients when they are hospitalized in intensive care with parenteral nutrition and if they don’t receive parenteral nutrition, the can’t cause bloodstream infections? If no, please correct the meaning of the sentence.

  • It is true that these fungi can cause bloodstream infections in these patients when they are on lipid parenteral nutrition. To our knowledge, cases of bloodstream infection outside of lipid parenteral nutrition are very rare.

Line 35: Again, what is Dixon? Is it liquid or solid?

  • Dixon is a culture medium for Malassezia It is a solid medium, but in our study we prepared it as a liquid medium (without agar).

Line 36: Please write microorganisms in italic throughout the whole manuscript.

  • See corrections

Line 37: Candida spp. are yeasts. Candida spp. and moulds or yeasts in including Candida spp. and moulds?

  • It is well noted............ variety of fungi including Candida and molds. Candida spp. are yeast, but molds are not yeast.

Line 39. Please delete the word yeast.

  • See correction line 39

Line 40. Should read “seem”.

  • "seems" is correct in the sentence (the use of....... seems). line 43

Line 43. Should read “cycloheximide”.

  • See correction line 44

Line 51: This “is” a …, Streptomyces hygroscopicus “and” has.. Please let the English of this manuscript be corrected by native English speaker. I stop correcting errors at this point.

  • See correction line 53-54

Materials and Methods

Line 64: How is it possible if you have 3 strains from 3 different species to do susceptibility testing of rapamycin on 2 M. furfur strains?

  • We had 3 Malassezia species ( furfur, M. sympodialis and M. pachydermatis). The susceptibility testing was performed on 2 strains of M. furfur. (See line 66)

Line 67: What means maintained? you stored them at 30°C in this medium? How where the strains subcultured before the experiments? Time of growth? Growth conditions? How did you culture the Candida strains?

  • “Maintained” means cultured. This is a term commonly used in these contexts for susceptibility testing of strains. (See correction section 2.1)

Line 74/76: Why do you change the unit from mg/ml to mg/L?

  • The mg/mL was the unit of rapamycin stock solution provided by the supplier (Sigma). The mg/L or µg/mL are the international units used in MIC tests. It was more convenient for us to change from mg/ml to mg/L.

Line 75-76: It is more than unclear how these dilutions have been performed. If rapamycin at a concentration of 1000µg/ml in cytotoxic acetonitrile was used, the final concentration of acetonitrile in the well containing 32µg/ml would be >3%!! This is way too much to run to run susceptibility testing without cytotoxicity of acetonitrile. Why didn’t you use DMSO as a solvent? Up to 200mg are soluble per ml. With DMSO it would easily be possible to stay under the 0.5% v/v final concentration of DMSO in each well.

  • The dilutions were made in a cascade of 1/2 from the stock solution.

For example: prepare a solution of rapamycin at 64 mg/L in Dixon liquid medium and then make cascade dilutions (64 - 0.125 mg/L). Then add the inocula to have final concentrations of 32 - 0.0625 mg/L.

  • Acetonitrile did not affect the growth of yeasts tested, as both Malassezia and krusei strains grew in all wells. Again, rapamycin was provided in solution in acetonitrile by the supplier (Sigma). We could not use DMSO because it was not a powdered molecule.

It is not clear if all wells contained to same concentration of acetonitrile, including the growth control.

  • See above

There exist protocols of modified EUCAST susceptibly testing of Malassezia spp. (10.1128/JCM.00338-17) which could have been used.

  • This is what we used, the modified EUCAST protocol. It should be noted that there is no standardized protocol for Malassezia.

Line 80: Standardized to what? 0.5 McFarland?

  • The standardization was performed using the densitometer at 0.5 McFarland. (See line 83)

Line 87-89: How is it possible to read an inhibition of 90% by eye? When I read MIC with 90% of inhibition I use a photometer.

  • To read the MICs, we used inclined mirrors that allow a closer view of the wells of each plate.

Line 94: How much rapamycin was added to the medium?

  • See above

Line 95: Did all petri dishes contain the same amount of medium?

  • Yes, all plates contained the same amount of medium (approximately 20 mL).

Results

Line 104-105: Did you use the same concentrations of acetonitrile in the other paper?

  • This is the first paper where we performed the test and there is no other paper.

Line 108: It’s recommended to read MIC of Candida species after 24h of incubation.

  • Yes, it's true and that's what we did. We also read at 48 h to compare 24 h as we had read the Malassezia MICs at 48 h.

Line 109-110: Required 48h of reading time or growth time?

  • The 48 h is the minimum time used for reading Malassezia MICs, as these yeast grow slowly.

Line 111: That’s wrong. It is not suggesting that rapamycin has no activity against Malassezia spp. as you only tested 2 M. furfur strains! What about the other 17 species from the genus Malassezia? Why were the other species of your collection not tested?

  • It was reported that furfur and M. sympodialis (as these are the species tested) are not sensitive to rapamycin (see reference 29). Here we tested 2 M. furfur as proof of concept in liquid medium. Afterwards, we tested on solid medium with 3 species and found no effect of rapamycin on the tested species. We did not test several species because we did not have all Malassezia species. Moreover, we had an ongoing study on the isolation of Malassezia from the skin with this medium, and so far 5 species have been isolated, including M. furfur, M. sympodialis, M. globosa, M. restricta and M. dermatis.

Line 119: By testing 1 stain of C. albicans and 1 C. tropicalis you did not show that most strains of this species are susceptible to rapamycin.

  • The effect of rapamycin against Candida was already known in the literature (see reference: 25, 27 and 28). Therefore, we did not need to test many strains.

Discussion

Line 133: Issue or challenge?

  • Challenge (see correction line 136)

Line 137: Limited to 2 tested isolates of M. furfur.

  • See correction line 139-140

Line 157: These results shows that rapamycin could be.

  • corrected

Line 159-161: When have you done this? I have only seen incubation with Candida spp. or Malassezia spp.

  • This was a reminder from the literature (reference 28) to show the importance of using rapamycin.

Line 163-166: You can propose to use your medium on polymicrobial samples, but you have not shown that it works. To my opinion your results warrant further investigation on polymicrobial samples, after it has been tested if the other clinically relevant Malassezia have been tested for their susceptibility to rapamycin.

  • More in-depth studies are in progress in other projects.

Conclusion

Please delete conclusion or line 163-166 of the discussion.

  • Sentences deleted.

Reviewer 2

Comments and Suggestions for Authors

Comments:

Authors can describe in detail Fas Fung medium:

  • See line 70-72

Bittar F, Gouriet F, Khelaifia S, Raoult D, Ranque S. FastFung: A novel medium for the culture and isolation of fastidious fungal species from clinical samples. J Microbiol Methods. 2021 Jan;180:106108. doi: 10.1016/j.mimet.2020.106108. Epub 2020 Nov 21. PMID: 33232796.

FastFung medium is suitable for the culture of clinical fungi, including fastidious ones, for both research and diagnostic studies. It is based on Schaedler agar supplemented with many essential components for the growth of fastidious fungi. It also contains selective antibacterial agents for the inhibition of contaminant bacteria growth. In this preliminary study, the FastFung medium was compared to the gold standard Sabouraud medium for 98 fungal and 20 bacterial strains. The fungal strain positive culture rate was 100% vs. 95% and the bacterial strain inhibition was 100% vs. 20%, for the FastFung and Sabouraud media, respectively. When compared to the Sabouraud medium on 120 clinical samples, the FastFung medium displayed both a higher fungal colonies count, and a lower culture contamination rate. Storage at 4°C for 4 weeks did not alter the FastFung culture medium performances for the six isolates of Candida, Cryptococcus, and Penicillium tested. These encouraging results suggest future development of using the FastFung medium in clinical mycology and in mycobiome characterization. Further prospective evaluation aiming at assessing whether implementing the FastFung medium in the routine workflow simplifies and strengthen fungal isolation capacities in the clinical laboratory is warranted.

Authors can mention, that Malassezia play important role in atopic dermatitis patients.

  • See line 32-33 (reference 7)

I suggest you can cite:

 Celakovska J, Vankova R, Bukac J, Cermakova E, Andrys C, Krejsek J. Atopic Dermatitis and Sensitisation to Molecular Components of Alternaria, Cladosporium, Penicillium, Aspergillus, and Malassezia-Results of Allergy Explorer ALEX 2. J Fungi (Basel). 2021 Mar 4;7(3):183. doi: 10.3390/jof7030183. PMID: 33806376; PMCID: PMC8001933.

Reviewer 3

Comments and Suggestions for Authors

“The authors of the MS entitled “MalaSelect: a selective culture medium for Malassezia species isolation” provide information of a potential isolation medium for these fastidious species.

It’s a good idea, but the MS needs English grammar and scientific terminology editing and could be shortened (too long for the data presented). Below are some edits that can help the revision of the article.

Title, delete the word “isolation”. Is the word “selective” needed or should it be placed after medium and between parenthesis?

  • See correction (title)

Please reword, the first two sentences; also what about bacterial isolates?

  • The question addressed here was the inhibition of non-Malassezia yeasts, which pose a problem in culture. For bacteria, there are many antibiotics such as chloramphenicol etc. that can be added to the medium.

“Including C. albicans…”.The reminding text is not needed.   It “does not resist”, it is resistant…..

  •  

Line 38-39: “including Candida spp. yeast and moulds [10–13]. “Relatively slow growing Malassezia spp. yeast” . Yeast is redundant since both Candida and Mal.. are yeasts.

  • See correction line 38-40

Line 41. “Ineffective” is not the appropriate term here. The issue is that overgrowth of other microorganisms precludes the growth/isolation of these more fastidious species.

  • See correction line 42-43

Line 53: is this statement correct? Has it been approved?

  • Yes, rapamycin has antifungal activity, and this has been shown for Candia albicans (reference 25, 27 and 28).

Line 71, the word “strain” is not needed.

  • See correction line 71

Lines 78-82. Provide only the required information, the steps for inoculum preparation are provided in both CLSI and EUCAST docs; perhaps a ref. here?

  • It is necessary to detail this to avoid confusion and misunderstanding by some readers (see line 86-87).

Line 88. The term “producing” is not the proper here.

  • See correction line 92-94

Lines 91-94: Again, redundant information.

  • See correction line 91-92

Line 123: Should read better: Rapamycin MICs for isolates of M. furfur spp. and Candida spp.

Is there a need for the figures? They are confusing.

  • See correction line 123

Lines 133-35. The term “issue” is not redundant. The authors mean perhaps: The supplement with ant…

The discussion and certainly the conclusions should be shortened.

  • See correction

Reviewer 2 Report

Comments:

Authors can describe in detail Fas Fung medium:

Bittar F, Gouriet F, Khelaifia S, Raoult D, Ranque S. FastFung: A novel medium for the culture and isolation of fastidious fungal species from clinical samples. J Microbiol Methods. 2021 Jan;180:106108. doi: 10.1016/j.mimet.2020.106108. Epub 2020 Nov 21. PMID: 33232796.

FastFung medium as suitable for the culture of clinical fungi, including fastidious ones, for both research and diagnostic studies. It is based on Schædler agar supplemented with many essential components for the growth of fastidious fungi. It also contains selective antibacterial agents for the inhibition of contaminant bacteria growth. In this preliminary study, the FastFung medium was compared to the gold standard Sabouraud medium for 98 fungal and 20 bacterial strains. The fungal strain positive culture rate was 100% vs. 95% and the bacterial strain inhibition was 100% vs. 20%, for the FastFung and Sabouraud media, respectively. When compared to the Sabouraud medium on 120 clinical samples, the FastFung medium displayed both a higher fungal colonies count, and a lower culture contamination rate. Storage at 4 °C for 4 weeks did not alter the FastFung culture medium performances for the six isolates of Candida, Cryptococcus, and Penicillium tested. These encouraging results suggest future development of using the FastFung medium in clinical mycology and in mycobiome characterization. Further prospective evaluation aiming at assessing whether implementing the FastFung medium in the routine workflow simplifies and strengthen fungal isolation capacities in the clinical laboratory is warranted.

Authors can mention, that Malassezia play important role in atopic dermatitis patients.

I suggest you can cite:

 Celakovska J, Vankova R, Bukac J, Cermakova E, Andrys C, Krejsek J. Atopic Dermatitis and Sensitisation to Molecular Components of Alternaria, Cladosporium, Penicillium, Aspergillus, and Malassezia-Results of Allergy Explorer ALEX 2. J Fungi (Basel). 2021 Mar 4;7(3):183. doi: 10.3390/jof7030183. PMID: 33806376; PMCID: PMC8001933.

Author Response

Authors can describe in detail Fas Fung medium:

  • See line 70-72

Bittar F, Gouriet F, Khelaifia S, Raoult D, Ranque S. FastFung: A novel medium for the culture and isolation of fastidious fungal species from clinical samples. J Microbiol Methods. 2021 Jan;180:106108. doi: 10.1016/j.mimet.2020.106108. Epub 2020 Nov 21. PMID: 33232796.

FastFung medium is suitable for the culture of clinical fungi, including fastidious ones, for both research and diagnostic studies. It is based on Schaedler agar supplemented with many essential components for the growth of fastidious fungi. It also contains selective antibacterial agents for the inhibition of contaminant bacteria growth. In this preliminary study, the FastFung medium was compared to the gold standard Sabouraud medium for 98 fungal and 20 bacterial strains. The fungal strain positive culture rate was 100% vs. 95% and the bacterial strain inhibition was 100% vs. 20%, for the FastFung and Sabouraud media, respectively. When compared to the Sabouraud medium on 120 clinical samples, the FastFung medium displayed both a higher fungal colonies count, and a lower culture contamination rate. Storage at 4°C for 4 weeks did not alter the FastFung culture medium performances for the six isolates of Candida, Cryptococcus, and Penicillium tested. These encouraging results suggest future development of using the FastFung medium in clinical mycology and in mycobiome characterization. Further prospective evaluation aiming at assessing whether implementing the FastFung medium in the routine workflow simplifies and strengthen fungal isolation capacities in the clinical laboratory is warranted.

Authors can mention, that Malassezia play important role in atopic dermatitis patients.

  • See line 32-33 (reference 7)

I suggest you can cite:

 Celakovska J, Vankova R, Bukac J, Cermakova E, Andrys C, Krejsek J. Atopic Dermatitis and Sensitisation to Molecular Components of Alternaria, Cladosporium, Penicillium, Aspergillus, and Malassezia-Results of Allergy Explorer ALEX 2. J Fungi (Basel). 2021 Mar 4;7(3):183. doi: 10.3390/jof7030183. PMID: 33806376; PMCID: PMC8001933.

Reviewer 3 Report

“The authors of the MS entitled “MalaSelect: a selective culture medium for Malassezia species isolation” provide information of a potential isolation medium for these fastidious species.

It’s a good idea, but the MS needs English grammar and scientific terminology editing and could be shortened (too long for the data presented). Below are some edits that can help the revision of the article.

  1. Title, delete the word “isolation”. Is the word “selective” needed or should it be placed after medium and between parenthesis?
  2. Please reword, the first two sentences; also what about bacterial isolates?
  3. “Including C. albicans…”.The reminding text is not needed.   It “does not resist”, it is resistant…..

Line 38-39: “including Candida spp. yeast and moulds [10–13]. “Relatively slow growing Malassezia spp. yeast” . Yeast is redundant since both Candida and Mal.. are yeasts.

Line 41. “Ineffective” is not the appropriate term here. The issue is that overgrowth of other microorganisms precludes the growth/isolation of these more fastidious species.

Line 53: is this statement correct? Has it been approved?

Line 71, the word “strain” is not needed.

Lines 78-82. Provide only the required information, the steps for inoculum preparation are provided in both CLSI and EUCAST docs; perhaps a ref. here?

Line 88. The term “producing” is not the proper here.

Lines 91-94: Again, redundant information.

Line 123: Should read better: Rapamycin MICs for isolates of M. furfur spp. and Candida spp.

Is there a need for the figures? They are confusing.

Lines 133-35. The term “issue” is not redundant. The authors mean perhaps: The supplement with ant…

The discussion and certainly the conclusions should be shortened.

Author Response

“The authors of the MS entitled “MalaSelect: a selective culture medium for Malassezia species isolation” provide information of a potential isolation medium for these fastidious species. It’s a good idea, but the MS needs English grammar and scientific terminology editing and could be shortened (too long for the data presented). Below are some edits that can help the revision of the article.

Title, delete the word “isolation”. Is the word “selective” needed or should it be placed after medium and between parenthesis?

  • See correction (title)

Please reword, the first two sentences; also what about bacterial isolates?

  • The question addressed here was the inhibition of non-Malassezia yeasts, which pose a problem in culture. For bacteria, there are many antibiotics such as chloramphenicol etc. that can be added to the medium.

“Including C. albicans…”.The reminding text is not needed.   It “does not resist”, it is resistant…..

  •  

Line 38-39: “including Candida spp. yeast and moulds [10–13]. “Relatively slow growing Malassezia spp. yeast” . Yeast is redundant since both Candida and Mal.. are yeasts.

  • See correction line 38-40

Line 41. “Ineffective” is not the appropriate term here. The issue is that overgrowth of other microorganisms precludes the growth/isolation of these more fastidious species.

  • See correction line 42-43

Line 53: is this statement correct? Has it been approved?

  • Yes, rapamycin has antifungal activity, and this has been shown for Candia albicans (reference 25, 27 and 28).

Line 71, the word “strain” is not needed.

  • See correction line 71

Lines 78-82. Provide only the required information, the steps for inoculum preparation are provided in both CLSI and EUCAST docs; perhaps a ref. here?

  • It is necessary to detail this to avoid confusion and misunderstanding by some readers (see line 86-87).

Line 88. The term “producing” is not the proper here.

  • See correction line 92-94

Lines 91-94: Again, redundant information.

  • See correction line 91-92

Line 123: Should read better: Rapamycin MICs for isolates of M. furfur spp. and Candida spp.

Is there a need for the figures? They are confusing.

  • See correction line 123

Lines 133-35. The term “issue” is not redundant. The authors mean perhaps: The supplement with ant…

The discussion and certainly the conclusions should be shortened.

  • See correction

Round 2

Reviewer 1 Report

Dear authors,

Although mistakes have been corrected, I still have to renew my concerns:

  1. Susceptibility testing has not been performed how it should have been done. Yes, there exists no standardization for susceptibility testing of Malassezia, but the authors claim to use a modified EUCAST protocol. In the EUCAST protocol the concentration of the toxic compound/solvent (in this case acetonitrile) is the same in every well. Here, the authors make a dilution to 64 µg/ml and then make dilutions by ½. This leads to different concentrations of acetonitrile in every well. This is a technical error! This problem could have been elimated if rapamycin would have been bought as powder and the dilutions would have been performed like in the EUCAST (or NSCLI) protocol (leads to same amount of solvent in every well).

  1. Only a very limited number of strains have been tested. It has not been shown that the medium works for other strains or species, too.

 Additionally, lines 158-159: I already know that not all molds, in this case the Mucorales, are susceptible to Rapamycin (Sirolimus). Schwarz et al, 2019, Journal of Antimicrobial Chemotherapy.

  1. To my opinion the conclusion is not supported by the results. The authors could say that their results warrant further investigation on polymicrobial samples, but they propose to use their medium for tests on polymicrobial samples (as they say based on their findings), but without have tested if it works!

Author Response

As required by the editor, we have added: "There is no standardization for susceptibility testing of Malassezia, and only a very limited number of strains have been tested, so this result warrant further investigation." in the conclusion section.

.